# Gibberellic Acid Modifies the Transcript Abundance of ABA Pathway Orthologs and Modulates Sweet Cherry (*Prunus avium*) Fruit Ripening in Early- and Mid-Season Varieties

**DOI:** 10.3390/plants9121796

**Published:** 2020-12-18

**Authors:** Nathalie Kuhn, Claudio Ponce, Macarena Arellano, Alson Time, Boris Sagredo, José Manuel Donoso, Lee A. Meisel

**Affiliations:** 1Facultad de Ciencias Agronómicas y de los Alimentos, Pontificia Universidad Católica de Valparaíso, Valparaíso 2340025, Chile; nathalie.kuhn@pucv.cl; 2Instituto de Tecnología de los Alimentos (INTA), Universidad de Chile, El Líbano 5524, Macul 7830490, Chile; claudio.ponce@ug.uchile.cl (C.P.); macarena.arellano@inta.uchile.cl (M.A.); atime007@ug.uchile.cl (A.T.); 3Programa de Doctorado en Ciencias Silvoagropecuarias y Veterinarias, Campus Sur Universidad de Chile, Santa Rosa 11315, La Pintana, Santiago 8820808, Chile; 4Instituto de Investigaciones Agropecuarias, INIA Rayentué, Av. Salamanca s/n Sector Los Choapinos, Rengo 2940000, Chile; bsagredo@inia.cl (B.S.); jdonoso@inia.cl (J.M.D.)

**Keywords:** abscisic acid, GA_3_, gibberellin, IAD, fruit tree management, non-climacteric, ripening, sweet cherry

## Abstract

Several phytohormones modulate ripening in non-climacteric fruits, which is triggered by abscisic acid (ABA). Gibberellins (GAs) are present during the onset of ripening in sweet cherry fruits, and exogenous gibberellic acid (GA_3_) application delays ripening, though this effect is variety-dependent. Although an ABA accumulation delay has been reported following GA_3_ treatment, the mechanism by which GA modulates this process has not been investigated at the molecular level in sweet cherry. Therefore, the aim of this work is to analyze the effect of GA_3_ on the fruit ripening process and the transcript levels of ABA pathway orthologs in two varieties having different maturity time phenotypes. The early-season variety had a rapid transition from yellow to pink fruit color, whereas pink color initiation took longer in the mid-season variety. GA_3_ increased the proportion of lighter colored fruits at ripeness in both varieties, but it produced a delay in IAD—a ripening index—only in the mid-season variety. This delay was accompanied by an increased transcript abundance of *PavPP2Cs*, which are putative negative regulators of the ABA pathway. On the other hand, the early-season variety had increased expression of *PavCYP707A2*—a putative ABA catabolic gene–, and reduced transcript levels of *PavPP2Cs* and *SnRK2s* after the GA_3_ treatment. Together these results show that GA modulates fruit ripening, exerting its action in part by interacting with the ABA pathway in sweet cherry.

## 1. Introduction

The ripening process of fleshy fruits is the sum of events, including softening, color change, decreased acidity, and sweetening that lead to maturity or ripeness when the fruit is suitable to be eaten [1]. In this regard, ripening is different from senescence processes in which the senescent tissues do not recover metabolites; instead, different sugars accumulate to high levels in the fruits [2]. Ripening is preceded by fruit degreening and reactivation of growth occurring at the onset of Stage III [1]. Other changes occurring during the ripening process include chloroplasts conversion into pigment accumulating chromoplasts, cell wall modification leading to a decline in firmness, and anthocyanin production, especially in the fruit skin [3].

Molecular, cellular, and physiological changes occurring during ripening are controlled by plant hormones that regulate this process [4,5,6]. In non-climacteric fruits, whose maturation occurs exclusively on the tree without dependence on ethylene production, several phytohormones are involved in the control of ripening [7,8,9]. Abscisic acid (ABA) plays a central role as it triggers color change in red and purple fruits, increases anthocyanin and sugar content, and reduces fruit firmness and acidity [10,11,12,13,14]. Ethylene may have a promoting effect, though it is minor compared with climacteric fruits, possibly through interaction with the ABA pathway [13,15]. Other phytohormones, including auxins, gibberellins, cytokinins, and brassinosteroids, are also able to modify ripening parameters when they are exogenously applied [16,17,18,19,20]. These phytohormones show content variations during ripening [19,20,21,22,23], and some of them act by modifying ABA levels [17,20,24] and producing a fruit ripening delay [24,25,26]. In this regard, a negative correlation has been reported between the gibberellin GA_4_ and some ripening parameters at harvest [23].

Gibberellins (GAs) are acidic diterpenoids involved in regulating several developmental processes [27,28]. Despite the widespread agricultural practice of gibberellic acid (GA_3_) applications to modify fruit parameters, the role of GA in the ripening process has not been clearly elucidated in non-climacteric fruits. GA and ABA transcriptionally antagonize each other during plant development, and GA negatively regulates ABA biosynthesis, perception, and signal transduction [29,30]. In sweet cherry (*Prunus avium* L.) fruits, GA_3_ treatment delays the ABA accumulation at the onset of ripening [17].

Sweet cherry is a non-climacteric fleshy fruit plant with a typical double-sigmoid growth pattern characterized by fast growth occurring at Stage I; slow growth of the exocarp and mesocarp, embryo development and endocarp lignification leading to pit hardening in Stage II; and growth resumption and ripening initiation at Stage III [31,32,33,34]. However, unlike other non-climacteric fruit species, *Prunus avium* has a very short fruit ripening period. During the sweet cherry fruit development, there is a transition from green fruits to yellow, straw-yellow, pink, and red. Sweet cherry pink color initiation and sugar increase at the onset of ripening correlate with ABA accumulation, which is known to trigger ripening [10,13,35]. Other phytohormones’ role in the control of sweet cherry fruit ripening has not been thoroughly investigated; however, there is evidence on the effect of GA_3_ in agronomic practices, where the treatment with GA_3_ at Stage II increases fruit firmness and size [36,37,38] and delays ripening by several days [36,39,40]. Usenik et al. [40] reported a delay in color development in GA_3_-treated fruits, which was detected as early as one-week posttreatment. A delay in sugar increase has also been reported [41]. Different effects of GA_3_ applications have been reported in sweet cherry varieties depending on whether they are early-season or mid-season maturing, which differ in the extension of Stage II, where pit hardening occurs [42]. Early-season maturing varieties have been reported as unaffected in fruit size after GA_3_ treatment, while mid-season varieties show delayed fruit size increase [36]. Additionally, polygalacturonase and Cx-cellulase activity are also delayed in mid-season varieties, whereas early-season varieties presented a trend towards more enzyme activity at harvest time, though not significant [36]. Regarding fruit color, GA_3_ treatment produces a delay in color development in mid-season varieties [40], while in an early-season variety, less—but not delayed—anthocyanin accumulation is observed upon GA_3_ application [17].

GAs are present at the onset of sweet cherry fruit ripening. At the beginning of Stage III, there is a peak in GA activity in the seed, while the maximum GA activity in the pericarp is observed a few days later [43]. Teribia et al. [23] found that gibberellin A_4_ (GA_4_) and GA_3_ were high at the onset of sweet cherry color initiation but then decreased as the fruits became darker, while gibberellin A_7_ (GA_7_) and gibberellin A_1_ (GA_1_) were constant during ripening. The molecular mechanisms by which GA controls fruit ripening are poorly understood. In grapevine (*Vitis vinifera* L.), PP2Cs and SnRK2s ABA pathway orthologs express during fruit ripening [44], and GA_3_ treatment modifies the expression of PP2Cs [45], which code for putative negative regulators of ABA pathway. In sweet cherry, orthologs of ABA signaling pathway, *PP2Cs* and *SnRKs*, express during fruit ripening [20,46], though their modulation by gibberellin at the onset of fruit ripening had not been evaluated in sweet cherry fruits. Therefore, this work aims to determine if the physiological effects produced by GA_3_ are associated with changes in transcript accumulation of ABA pathway-related genes. For this, we analyzed the effect of GA_3_ on transcript abundance of putative ABA homeostasis genes, *PavNCED1* and *PavCYP707A2*, previously involved in the regulation of ABA levels in sweet cherry [47,48], and on the transcript accumulation of the putative ABA signaling pathway genes, subfamily 2 of SNF1-related kinases (*PavSnRK2s*) and type 2C protein phosphatases (*PavPP2Cs*), expressed during sweet cherry fruit development [20,46]. The use of early- and mid-season varieties provides insight into how GA_3_ acts at the physiological and molecular level and sheds light on the role of GA during sweet cherry fruit ripening.

## 2. Results

### 2.1. Effect of GA_3_ on the Growth Curve of Early- and Mid-Season Sweet Cherry Varieties

To better appraise the physiological effects of GA_3_, we first characterized fruit growth dynamics along with phenology. Fruit growth, as width variations, was described by a double sigmoidal curve in both varieties (Figure 1), though a more extended slow growth period and a longer light green-to-yellow transition was observed in the mid-season variety, together with an additional growth peak a few days before harvest (Figure 1B). From the onset of color initiation to harvest, a similar time frame was observed in both varieties (Figure 1). The transition from yellow to pink fruit color coincided with the maximum growth rate in the early-maturing variety, whereas in the mid-season variety, color change started several days after the fast growth initiation (Appendix A).

To describe different effects of GA_3_ prior to the onset of ripening, we performed exogenous GA_3_ treatment in early- and mid-season varieties at the light green-to-straw yellow transition stage of each genotype. The effect of the treatment on fruit growth was monitored during Phase III (Figure 2). GA_3_ treatment generated a significantly larger fruit width (Figure 2A) and weight (Table 1) at the end of the growing period in the early-season variety. This effect was observed a few days after GA_3_ was applied. In contrast, GA_3_ did not significantly change fruit width in the mid-season variety at harvest (Figure 2B), but did significantly change fruit weight (Table 1).

### 2.2. Effect of GA_3_ on Fruit Parameters and IAD Variations during Ripening

At harvest, both varieties had less color (Appendix A) and a lower proportion of fruits in the darker ‘3’ and ‘4’ color categories in the GA_3_-treated plants (Figure 3A,C). Additionally, GA_3_ application significantly increased fruit firmness in both varieties at harvest (Table 1).

We aimed to detect the moment that GA_3_ starts to exert its action in early- and mid-season varieties; therefore, we measured the index of absorbance difference (IAD), which is a maturity index used for nondestructive assessment of the progression of the ripening process [49]. IAD indicates the presence of phenolic compounds that filter the chlorophyll absorbance and significantly correlates with anthocyanin content in sweet cherry ripened fruits [50]. GA_3_ affected the IAD of both varieties in the two seasons analyzed (Figure 3B; Appendix A). In the early-season variety, GA_3_ significantly decreased IAD at the end of the ripening process (Figure 3B). In the mid-season variety, GA_3_ delayed IAD evolution since control fruits increased their IAD values before GA_3_-treated fruits (Figure 3D).

### 2.3. Transcript Variations of ABA Pathway Orthologs upon GA_3_ Treatment in Early- and Mid-Season Varieties

We investigated if the physiological effects of GA_3_ observed were accompanied by changes in the transcript abundance of ABA pathway orthologs that are expressed during sweet cherry fruit ripening. The variations in the transcript abundance of ABA homeostasis putative orthologs, *PavNCED1* and *PavCYP707A2*, were analyzed in response to GA_3_ treatment. These genes participate in regulating ABA content during sweet cherry fruit ripening [47,48]. *PavNCED1* codes for a putative 9-cis epoxycarotenoid dioxygenase, a key enzyme of the ABA biosynthetic route. *PavCYP707A2* codes for a putative ABA 8′-hydroxylase, whose ortholog in *Arabidopsis* is involved in ABA catabolism [51]. *PavNCED1* transcript accumulation closely reflects the sharp increase in ABA levels during the initiation of ripening in sweet cherry fruits, whereas *PavCYP707A2* decreases during this process [47,48,52].

Samples collected just before (T0) or five days after the GA_3_ treatment (T5) were analyzed. *PavNCED1* did not change significantly between GA_3_-treated and control fruits nor between T0 and T5 in both varieties (Figure 4). On the other hand, in the early-season variety, a five-fold increase in *PavCYP707A2* occurred five days after the GA_3_ treatment in relation to control fruits (Figure 4A).

We also investigated changes in the transcript accumulation of putative ABA signaling pathway genes, subfamily 2 of SNF1-related kinases (*PavSnRK2s*) and type 2C protein phosphatases (*PavPP2Cs*) upon GA_3_ treatment. These genes show variations during sweet cherry fruit development and are highly expressed during ripening [20,46], possibly participating in ABA signal transduction.

*PavSnRK2s* and *PavPP2Cs* transcripts accumulated from T0 to T5 and decreased upon GA_3_ treatment in the early-season variety, except for *PavSnRK2.1* (Figure 5A). In contrast, no significant differences in *PavSnRK2s* were observed from T0 to T5 in the mid-season variety, and *PavPP2Cs* transcripts upregulated after GA_3_ treatment (Figure 5B).

## 3. Discussion

### 3.1. Early- and Mid-Season Varieties Are Responsive to GA_3_

Stage II, which is characterized by slow growth, endocarp lignification, and embryo development, precedes the onset of ripening in sweet cherry fruits. Additionally, the most marked effect of GA_3_ on ripening delay occurs when applied at the end of Stage II [39], suggesting that this is a key moment for ripening initiation timing. Differences between early- and mid-varieties were related to this stage since the early-season variety had a shorter slow growth period than the mid-season variety (Figure 1). Variations in the duration of Stage II between varieties have been previously reported in fleshy fruits with double sigmoid growth patterns, such as grapevine and sweet cherry [1,36]. Possibly events that occur at the end of Stage II take more time to occur in mid- or late-varieties. In line with this idea, in the mid-season variety, we found that pink color initiation was delayed several days from growth resumption, whereas in the early-season variety, both events occurred simultaneously (Figure 1). A growth increase occurs at the onset of Stage III in many fleshy fruits due to variations in sink-source relations, cell wall modifications, and water inflow into the cell [1]. GA modifies cell wall composition during sweet cherry fruit ripening [17]. In line with this idea, Peréz et al. [53] proposed that increased cell wall invertase activity and hexose content could explain the berry-sizing effect of GA_3_ in grapevine. In agreement with this, we found that GA_3_ significantly increased the fruit width (Figure 2) and weight (Table 1) in the early-season variety at harvest, while the application of GA_3_ on the mid-season variety only caused a delay in fruit growth (Figure 2, Table 1). The delayed growth caused by GA_3_ may explain the fruit size reduction of GA_3_ in the mid-season variety. In this regard, we observed a slight delaying effect of GA_3_ on the size of this mid-season variety, whereby control fruits increased in size from 67 to 69, whereas GA_3_-treated fruits did not. Choi et al. [36] reported an increased size in Lapins fruit after GA_3_ treatment, but this is because they allowed the GA_3_-delated fruits to reach full maturity. Additionally, GA_3_ increased firmness at harvest in both varieties (Table 1). Choi et al. [36] found delayed Cx-cellulase and polygalacturonase activity upon GA_3_ treatment in mid-season variety and more—but not delayed—activity in an early-season variety.

Color initiation due to anthocyanin accumulation is controlled by ABA [14,52,54] but can be modulated by GAs, auxin, and cytokinins in non-climacteric species [16,17,18,19]. We found decreased but not delayed IAD values after GA_3_ treatment in the early-season variety at harvest (Figure 3B). In the mid-season variety, GA_3_ produced a significant delay in IAD evolution (Figure 3D). In both varieties, the effect on IAD was accompanied by an increased proportion of lighter fruits at the end of ripening (Figure 3A,C), showing a negative effect of GA_3_ on color development. In accordance with our results, color development was delayed by GA_3_ in mid-varieties [40]. An early-season variety showed decreased but not delayed anthocyanin accumulation [17].

The role of GAs has been poorly investigated in the context of non-climacteric fruit ripening. In grapevine, GA_3_ is not commonly used at *veraison* or ripening. Instead, it is applied before bloom to produce a thinning effect or after fruit set in seedless varieties for promoting cell enlargement [55]. GA_3_ delays ripening and produces cluster rigidity and pedicel thickening [56]. Therefore, in this model species, our understanding of the role of GAs during ripening is relatively poor as its agronomic use is not common. In the non-climacteric strawberry (*Fragaria* × *ananassa* Duch.), GA_3_ treatment during ripening delays anthocyanin accumulation and chlorophyll degradation [57]. GA_3_ and GA_4_ increase from green to red stages in strawberry fruits [58]. GAs distribution over time is quite similar to that found in sweet cherry [23]. Interestingly, GA_3_ and GA_4_ increases in strawberries are concomitant with increased *FaGID1c* expression from green to red stages, a putative receptor that immunolocalized in red fruits [58]. These results show that in this non-climacteric species, GA signaling is active during ripening. However, little is known about GA’s role in other non-climacteric fruit species, such as sweet cherry.

### 3.2. GA_3_ Treatment Affects the Transcript Abundance of ABA Pathway Orthologs in Early- and Mid-Season Varieties

ABA is the principal regulator of the fruit ripening process in the non-climacteric sweet cherry [10,35]. Recently, molecular and functional analyses of genes relevant for ABA homeostasis have been performed and found that *PavNCED1*, encoding a putative 9-cis-epoxycarotenoid dioxygenase, and *PavCYP707A2*, which codes for a putative ABA 8′-hydroxylase, regulate ABA levels during sweet cherry fruit ripening [13,47,48,52]. Here we investigated if *PavNCED1* and *PavCYP707A2* transcript abundance was modified upon GA_3_ treatment. GA_3_ significantly increased *PavCYP707A2* only in the early-season variety (Figure 4A), suggesting that GA could influence ABA degradation. In sweet cherry fruits, *PavCYP707A2* decreases as ABA increases [52], suggesting that less ABA degradation is necessary for its accumulation. RNAi lines with decreased expression of *PavCYP707A2* show altered expression of anthocyanin related genes involved in color initiation [47]. Previous reports found that this gene was upregulated by ABA in sweet cherry [48], but its regulation by other hormones has not been investigated to date. Here we show that GA could regulate *PavCYP707A2* transcript accumulation. This could eventually lead to decreased ABA levels, explaining less color observed in the early-season variety. In agreement with this, Kondo and Danjo [17] found that GA_3_ treatment delayed ABA accumulation in the fruits of the early-season variety Satonishiki.

*PavPP2Cs* and *PavSnRK2s* are putative negative and positive ABA signaling genes, respectively. In *Arabidopsis*, dephosphorylation of SnRKs by PP2Cs leads to repression of the ABA response, which is released when ABA binds to its receptor, which in turn interacts with PP2C, thus inhibiting its activity [59]. In sweet cherry, *PavPP2Cs* and *PavSnRK2s* transcript accumulation varies during fruit development and in response to ABA [20,46], with some being upregulated and others downregulated.

Here we studied *PavPP2C3*, *PavPP2C4*, *PavSnRK2.1*, *and PavSnRK2.3*, reported as highly expressed during sweet cherry fruit ripening [20], and found their transcript abundance was modulated by GA_3_. Regarding the hormonal regulation of these genes, the transcript abundance of *PavPP2C3*, *PavSnRK2.1* and *PavSnRK2.2* was decreased upon IAA treatment, while *PavPP2C4* was not affected [20]. In grapevine, downregulation of an ABA putative negative regulator, *PP2Cs*, was produced by GA_3_ treatment at fruit color change [45]. Therefore, it seems plausible that these genes are modulated by GA_3_ in sweet cherry.

In the early-season variety, GA_3_ inhibited the upregulation of ABA pathway-related genes from T0 to T5 (Figure 5A), showing that GA controls positive as well as negative ABA signaling genes. In contrast, in the mid-season variety, only the negative regulators, *PavPP2Cs*, increased upon the GA_3_ treatment (Figure 5B). We hypothesize that more negative regulator expression could cause a reduced ABA sensitivity, possibly leading to more ABA to be accumulated for obtaining the same ABA response, which could produce a delay in the occurrence of ripening related events. This hypothesis is consistent with our observation of IAD delay in mid-season variety upon GA_3_ treatment (Figure 3D). It also agrees with delayed Cx-cellulase activity reported by Choi et al. [36] and the delayed color development in other mid-season varieties [40].

In non-climacteric species, some forms of GA are present at the onset of fruit ripening [23,58]. GA perception and signaling components are expressed when strawberry fruits are red [58]. In grapevine, GA_3_ applied at *veraison* modifies ripening parameters and the expression of several genes involved in hormone biosynthesis and signaling, including *PP2Cs* [45]. In sweet cherry, GA_3_ treatment modifies the ABA accumulation pattern [17]. Finally, we show that GA modulates the transcript levels of some putative ABA homeostasis and signaling genes, therefore possibly crosstalk between GA and ABA exists at the onset of ripening in non-climacteric fruits. The possible role of GA in this process is discussed below.

### 3.3. Possible Role of GA_3_ during Fruit Ripening

The regulation of the ripening process in non-climacteric fruits has not been completely elucidated. Sweet cherry fruits are characterized as having a typical non-climacteric respiration pattern with little or null ethylene production [13,32,60]. In contrast, ABA is the phytohormone with the strongest effects on ripening triggering [10,13,35,48], and ethylene seems to play a minor role by influencing the effect of ABA [10]. Ethephon treatment induces *PavNCED1* expression and slightly increases ABA content [13].

Regarding the role of other hormones, our understanding is scarce. Wang et al. [20] reported a peak in the IAA content at the same time as ABA, and IAA treatment increased ABA content, though it did not affect anthocyanin concentration. Additionally, some components of the ABA signaling pathway were downregulated upon IAA treatment [20]. With respect to cytokinins, Teribia et al. [23] found a gradual increase in isopentenyladenosine (IPA) to reach high levels at the end of ripening and treatment with synthetic cytokinin affects color and increases the size and firmness of sweet cherry fruits [61]. Regarding GAs, GA_4_ and GA_3_ are high at the onset of color initiation [23] and GA_3_ treatment delays ripening [36,39,40], which also occurs in other non-climacteric species such as strawberries [57]. GA_3_ applications also reduce anthocyanin accumulation and modify cell wall composition in sweet cherry fruits [17].

GAs are present during sweet cherry fruit ripening [23,43]; however, GA_3_ applications are performed in the transition from light green to yellow [36,39]. On the other hand, usually, spatial separation of ABA and GA responses is controlled by hormone movement [62]. In light of this, it is possible to hypothesize that GAs biosynthesis occurs primarily in the seed, and then GAs are transported to the pericarp before endocarp lignification. This idea is supported by the fact that later applications of GA_3_, when the endocarp is fully lignified, do not produce a delaying effect [39]. Once in the mesocarp, GA may modulate the ABA pathway. This idea is consistent with studies in other species showing that GA and ABA are often spatially separated, as they play antagonistic roles in plant responses [30]. Therefore, such antagonism at the onset of fruit ripening between GA and ABA cannot be ruled out in sweet cherry fruit ripening.

The aim of this work was to determine if GA_3_ modulates fruit ripening in two sweet cherry varieties having contrasting maturity time phenotypes and if this was related to changes in the transcript abundance of ABA pathway-related genes. We found that both varieties were responsive to GA_3,_ and this was accompanied by changes in the transcript abundance of ABA homeostasis and signal transduction related genes. Further investigation is necessary to elucidate the different GA modes of action as well as the molecular features involved in the control of non-climacteric fruit ripening in order to optimize the timing of the application and use of this growth regulator in the field

## 4. Materials and Methods

### 4.1. Plant Material and GA_3_ Treatment

Fruits of *Prunus avium* L. varieties, Lapins and Glenred (Sequoia^®^), from five-year-old trees grafted on Cab-6P rootstocks growing in a commercial orchard located at Rengo, Chile (Long: O70°43′6.78″ Lat: S34°27′16.92″) were used in 2017–2018 and 2018–2019 seasons. Lapins is an early-flowering and mid-season maturing variety, whereas Glenred is an early-flowering and early-season maturing variety. Trees used in all the measurements were randomly selected from different rows of the orchard, and those plants from the beginning or the end of the row, less vigor or a visible difference in growth, were excluded from the analyses. Both varieties were characterized in their phenology throughout fruit development (Appendix A), according to Wenden et al. [63].

For the GA_3_ treatment, eight and ten trees of the Lapins and Glenred varieties, respectively, were randomly selected in the 2017–2018 and 2018–2019 seasons, and each tree was considered a biological replicate. GA_3_ (Activol^®^ 4%—Bayer Crop Science, Chile) was applied at a rate of 12.5 ppm to Lapins trees and at a rate of 18.6 ppm to Glenred trees. Untreated trees served as a control in Lapins and Glenred, respectively. GA_3_ was applied as a water solution with a hand sprayer until run-off at the light green-to-straw yellow transition stage of each variety (36 and 29 DAFB in Glenred and Lapins, respectively, in the 2017–2018 season; 38 and 35 DAFB in Glenred and Lapins, respectively, in the 2018–2019 season). For the RNA extractions, the T0 time point (minutes before the GA_3_ application) and T5 time point (5 days after the GA_3_ treatment) were utilized during the 2017–2018 season. All the fruit samples were collected in the field from three control and three GA_3_-treated plants. Samples were immediately frozen in liquid nitrogen.

### 4.2. Fruit Parameters

Fruit growth and IAD were measured in 20 fruits per tree every 2–5 days. A rank of 0 DAFB was assigned when 50% of the flowers were open. For fruit growth, the equatorial diameter was measured using a caliper.

Growth rate curves were generated using a slope calculation:
(*w*2 − *w*1)/(*t*2 − *t*1),
where *w* is fruit width, and *t* is time as DAFB. IAD (Cherry Meter (vis/NIR) device, T.R. ^®^ Turoni, Bologna, Italy) was measured in both cheeks of the fruits, and the average value was obtained during the 2017–2018 and 2018–2019 seasons. The values obtained from both cheeks of the fruits were averaged. For evaluations performed at harvest in the 2017–2018 season, fruits were collected at 59 and 69 DAFB in Glenred and Lapins, respectively. This date was determined according to sugar (14°–15°Bx) and firmness (min. 65 d.u.) agronomic maturity requirements. For color distribution of the fruits, soluble solids content, acidity, fruit firmness and weight, 25 fruits were randomly selected [64] from each of the control and GA_3_-treated trees. For color distribution, a CTIFL (Centre Technique Interprofessionnel des Fruits et Légumes) color chart from 1 to 4 was utilized (Appendix A). Weight was quantified in a portable mini scale. For firmness, the durometer (Durofel T.R. ^®^ Turoni, Forlì, Italy) was pressed on both sides of the fruits, giving a 1–100 value that was averaged. For soluble solids content and acidity determination (as malic acid%), five similar fruits among the 25 fruits were selected [64] and measured with a PAL-BX|ACID Pocket Sugar and Acidity Meter (ATAGO CO. LTD., Tokyo, Japan).

### 4.3. RNA Extraction and RT–qPCR Analysis

For RNA extractions, 0.5 g of fruit tissue (exocarp and mesocarp enriched) of each variety were ground using liquid nitrogen. Total RNA was isolated using the CTAB method with some modifications, as described in Meisel et al. [65]. Samples were treated with TURBO^TM^ DNase (Thermo Fisher Scientific, San Diego, CA, USA) according to the manufacturer’s instructions. Reverse transcription was performed using 1 µg RNA and the First-Strand cDNA synthesis system kit (Thermo Fisher Scientific, San Diego, CA, USA), following the manufacturer’s specifications. A260/230 and A260/A280 values were around 2.0 for all the samples. RNA electrophoresis of ribosomal bands using MOPS buffer was used to check RNA integrity (Appendix A).

Quantitative transcript abundance of sweet cherry was analyzed using real-time quantitative PCR (RT–qPCR) with two technical replicates and three biological replicates. Primers were selected from the literature [20,47,66,67] and tested for specificity (a single amplicon) by performing a dissociation curve (melting curve) analysis. When more than one amplicon was detected, primer sets were redesigned using the reported template sequence [20] and Primer-BLAST [68]. A single amplicon from these new primer sets was confirmed on a dissociation curve. The efficiency of these primers was determined by the LinRegPCR program [69]. Primer sequences are listed in Appendix A.

The RT–qPCR reaction was conducted on the QIAGEN Rotor-Gene Q system, using the Rotor-Gene Q Series software version 2.1.0. Fast Plus system. EvaGreen^®^ qPCR Master Mix (Biotium, Fremont, CA, USA) was used as an RNA intercalating agent, according to the manufacturer’s indications. RT–qPCR analyses were performed using MIQE guidelines for qRT-PCR [70] and the “golden rules of quantitative PCR” [71]. The primer efficiency was used for relative transcript abundance calculations, according to Pfaffi [72]. *PavCAC*, *PavTEF2*, *PavACT1* were used as reference genes. *PavCAC* was selected for the relative abundance calculations since it had less variation between samples (<1.5 Cq). All the genes analyzed had average Cq values higher than 28 cycles. Transcript abundance during development was set to 1.0 at T0. All the graphs were made with the GraphPad software Prism version 6.0e.

### 4.4. Statistical Analysis

The significance of variation between control and GA_3_-treated fruits at specific time points was tested by one-way ANOVA combined with Tukey’s post hoc analysis at *p* < 0.05.

## Figures and Tables

**Figure 1 plants-09-01796-f001:**
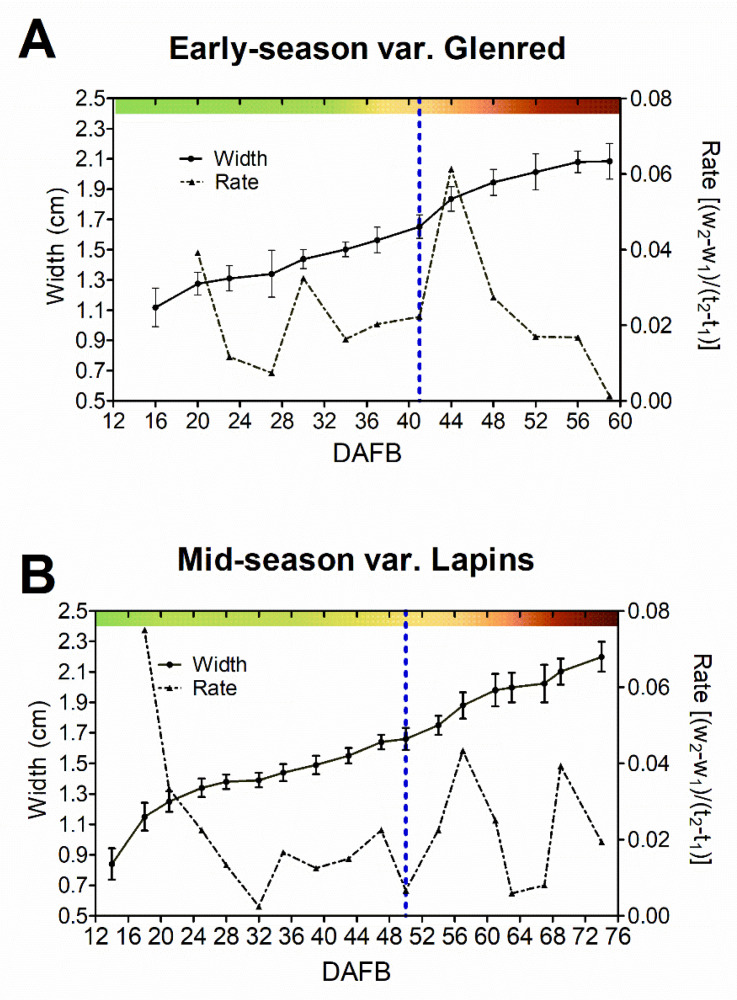
Fruit growth and growth rate curves of early-season variety, Glenred (**A**) and mid-season variety, Lapins (**B**). A color scale is included based on fruit phenology (Appendix A). Growth resumption prior to fruit color change is indicated with a dotted blue line, 20 fruits from Lapins and Glenred were randomly selected for measurements. Data as ± SEM. DAFB, days after full bloom.

**Figure 2 plants-09-01796-f002:**
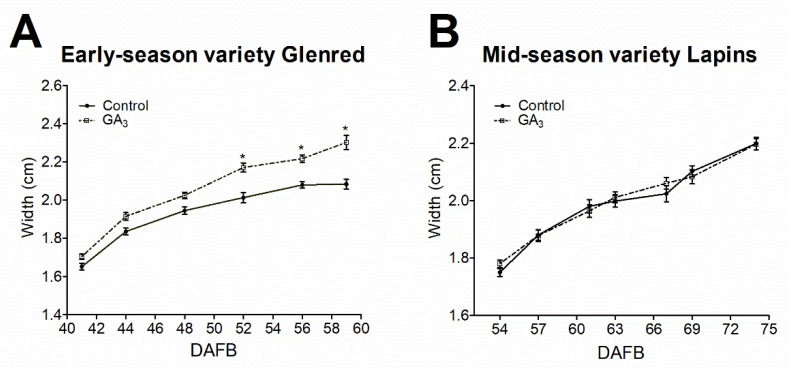
Effect of exogenous gibberellic acid (GA_3_) on fruit growth of early-season variety, Glenred (**A**) and mid-season variety, Lapins (**B**) 20 fruits from control and GA_3_ trees of Lapins and Glenred were randomly selected for measurements. Data as ± SEM ANOVA with Tukey’s post hoc test at *p* < 0.05 was conducted; “*” denotes statistical differences between GA_3_-treated and control fruits. DAFB, days after full bloom.

**Figure 3 plants-09-01796-f003:**
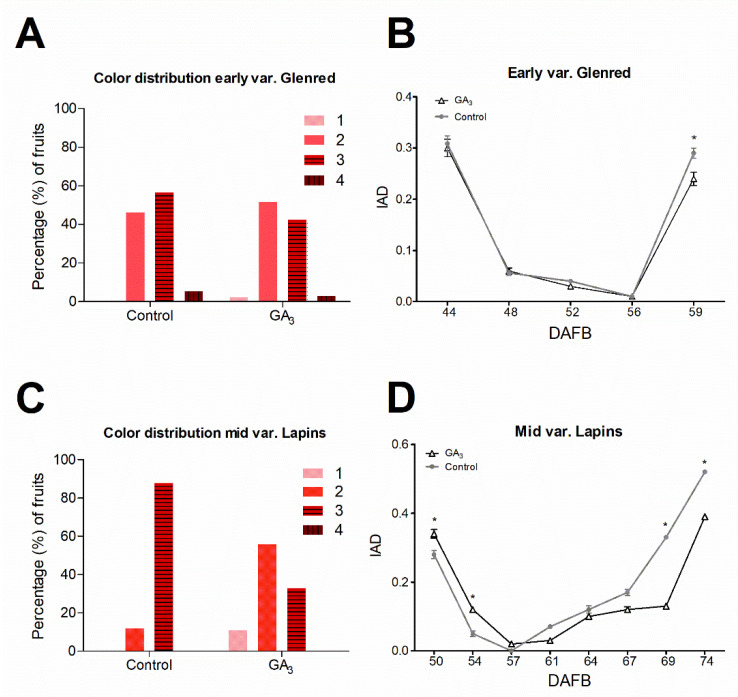
Effect of GA_3_ on color distribution and index of absorbance difference (IAD) at harvest of early-season variety, Glenred (**A**,**B**) and mid-season variety, Lapins (**C**,**D**). 20 fruits from control and GA_3_ trees of Lapins and Glenred were randomly selected for nondestructive IAD measurements in the field and 25 fruits for color distribution assessment at harvest. Data as ± SEM ANOVA with Tukey’s post hoc test at *p* < 0.05 was conducted; “*” denotes statistical differences between GA_3_-treated and control fruits CTIFL (Centre Technique Interprofessionnel des Fruits et Légumes) color chart was used in (**B**,**D**), where 1 is the lightest color, and 4 is the darkest color, respectively (Appendix A). DAFB, days after full bloom.

**Figure 4 plants-09-01796-f004:**
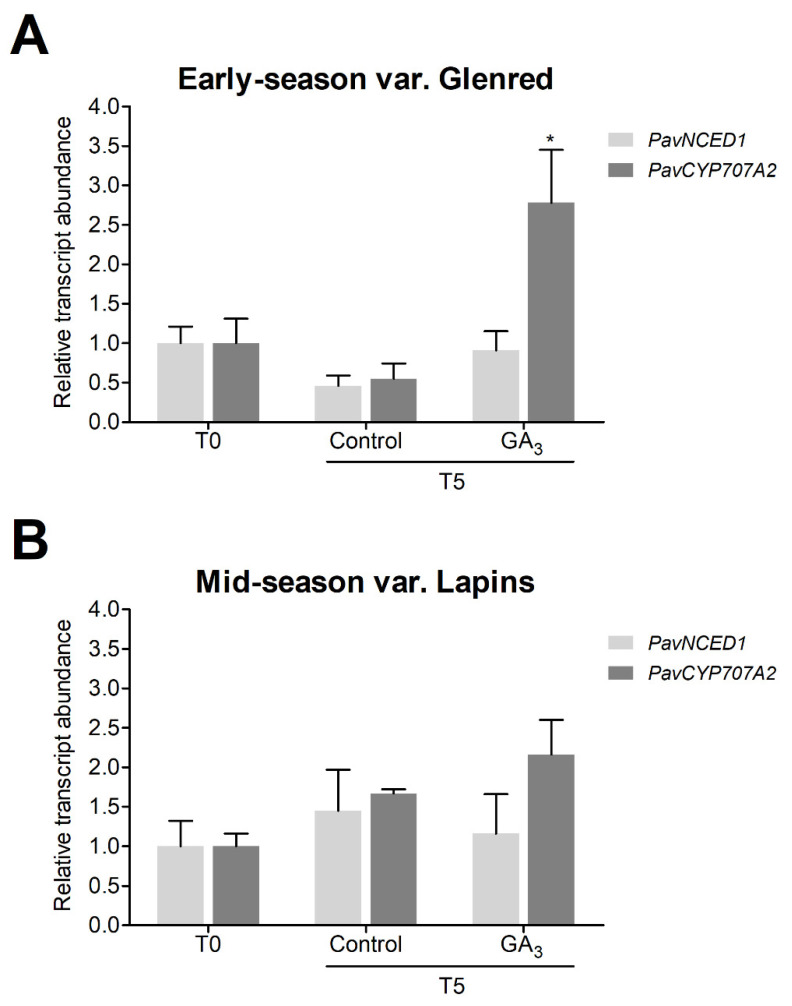
Effect of GA_3_ on transcript abundance relative to *PavCAC* of *PavNCED1* and *PavCYP707A2* sweet cherry orthologs, 5 days after the treatment (T5). Fruits from control and GA_3_ trees of (**A**) early-season variety Glenred and (**B**) mid-season variety Lapins, were randomly selected and pooled for the RT–qPCR analyses. Data as +SEM. Relative transcript abundance was set to 1.0 at T0, where T0 is the sampling performed immediately before GA_3_ application ANOVA with Tukey’s post hoc test at *p* < 0.05 was conducted; “*” denotes statistical differences between GA_3_-treated and control fruits for a gene.

**Figure 5 plants-09-01796-f005:**
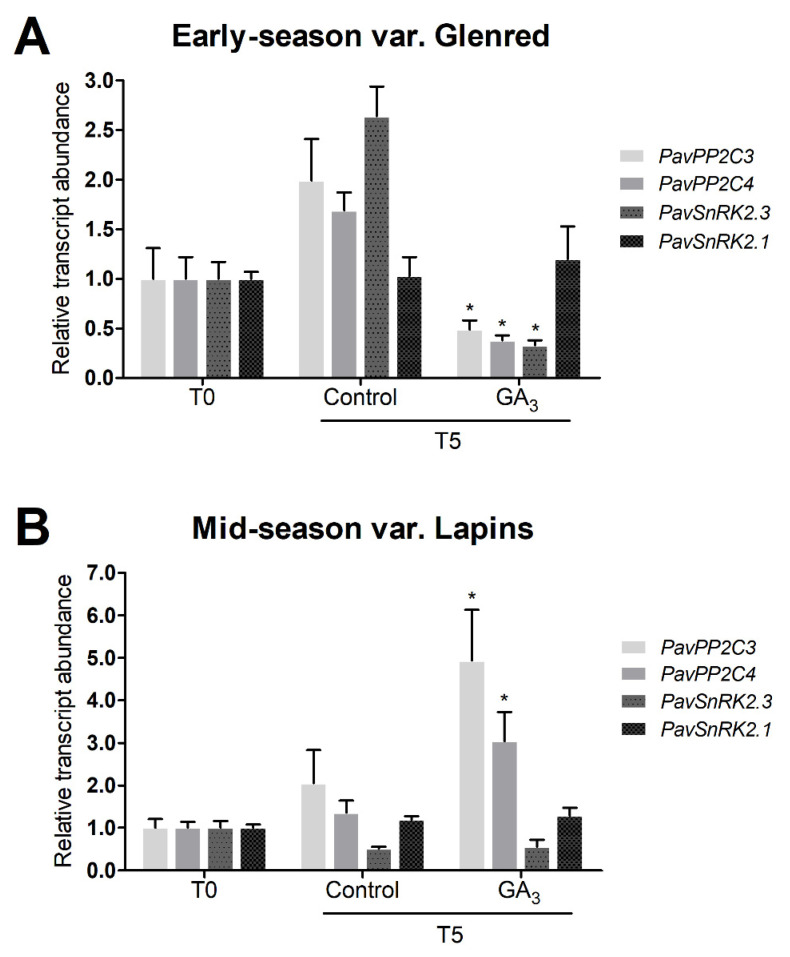
Effect of GA_3_ on transcript abundance relative to *PavCAC* of *PavPP2C* and *PavSnRK2* sweet cherry orthologs, 5 days after the treatment (T5). Fruits from control and GA_3_ trees of (**A**) early-season variety Glenred and (**B**) mid-season variety Lapins were randomly selected and pooled for the RT–qPCR analyses. Data as +SEM. Relative transcript abundance was set to 1.0 at T0, where T0 is the sampling performed immediately before GA_3_ application ANOVA with Tukey’s post hoc test at *p* < 0.05 was conducted; “*” denotes statistical differences between GA_3_-treated and control fruits for a gene.

**Table 1 plants-09-01796-t001:** Ripening parameters at harvest of mid- and early-season varieties in the 2017–2018 season. Mid-season variety harvest, 4 December 2017 (69 DAFB). Early-season variety harvest, 10 November 2017 (59 DAFB).

Variety	Treatment	Weight (g)	Firmness (du)	SSC (ºBrix)	Acidity (MA%)
Mid-season variety Lapins	Control	7.70b **	66.73a	14.64a	1.53a
	GA_3_ *	7.29a	68.09b	14.04a	1.43a
Early-season variety Glenred	Control	7.60a	73.29a	14.15a	1.43a
	GA_3_	8.29b	77.97b	14.60a	1.60a

* GA_3_ was applied as the commercial product Activol^®^ 4%—Bayer Crop Science, Chile to whole trees at the light green-to-straw yellow transition of fruits. ** The significance of variation between control- and GA_3_-treated fruits of each parameter was tested by one-way ANOVA analysis with Tukey’s post hoc test. Different letters denote significantly different means (*p* < 0.05). Lapins and Glenred were analyzed separately. SSC, soluble solid content; du, durometer units; MA malic acid.

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
