# Peer review of "Gibberellic Acid Modifies the Transcript Abundance of ABA Pathway Orthologs and Modulates Sweet Cherry (Prunus avium) Fruit Ripening in Early- and Mid-Season Varieties"

_plants, 2020, doi:10.3390/plants9121796_

Round 1

Reviewer 1 Report

A brief summary

The aim of this work was to analyze the effect of GA3 on the fruit ripening process and on the transcript levels of ABA pathway orthologs in two sweet cherries varieties. There has been already some studies about ripening and regulation by phytohormones in non-climacteric fruits published. Further research is, however, needed to better understand those complex mechanisms. Therefore, the topic of presented study is relevant and describes current problem.

Broad comments

  1. I found the paper to be overall important and describing a very interesting findings. The design of the experiment combined with phonological and molecular studies makes the dataset useful for the purpose.
  2. The manuscript is very well written, clear, precise, and easy to read.
  3. Authors mentioned RNA electrophoresis to check RNA integrity. I would appreciate if that gel picture was added as a supplementary material.
  4. The primers were selected from the literature, or re-designed from reported template sequence, using Primer-BLAST. Did authors check the efficiency of the primers and run a gel? Please, provide those pictures as a supplementary material.

Author Response

Point 1.1: I found the paper to be overall important and describing a very interesting findings. The design of the experiment combined with phonological and molecular studies makes the dataset useful for the purpose. The manuscript is very well written, clear, precise, and easy to read.

Response 1.1:  We would like to thank Reviewer 1 for their revision of the manuscript.

Point 1.2: Authors mentioned RNA electrophoresis to check RNA integrity. I would appreciate if that gel picture was added as a supplementary material.

Response 1.2: Gels confirming the integrity of the RNA have been added to the manuscript as Supplementary Figure 4.  The figure legend for this Supplementary Figure has been added to the paragraph entitled “Supplementary Materials” on lines 395-399. Additionally, this new supplementary figure has been mentioned in the Materials and Methods section on line 366 as well as the Supplementary Material doc file.

Point 1.3: The primers were selected from the literature, or re-designed from reported template sequence, using Primer-BLAST. Did authors check the efficiency of the primers?

Response 1.3:  The primers used in this work are listed in Table S3.  In order to determine primer specificity, a dissociation curve (melting curve) analysis was performed for all primer pairs.   Initially, primers were selected from the literature and all but two primer sets revealed a single peak (amplicon) in the dissociation curve.   The primer sets of PP2C3 and SnRK2.3 reported by Wang et al., 2015 revealed more than a single peak (amplicon).   Therefore, the target sequence describe by Wang et al 2015 was used to design more specific primer pairs using Primer-BLAST.  The newly designed primer sets mentioned in Table S3 and marked with an asterisk were analyzed for primer specificity by performing a dissociation curve, revealing a single peak (amplicon).  As mentioned on lines 370-371, “The efficiency of the primer sets were determined by the LinRegPCR program.”

In order to clarify these points better in the manuscript the following modifications were made to the text:

  • Lines 369-371 now state “Primers were selected from the literature [20,47,66,67] and tested for specificity (a single amplicon) by performing a dissociation curve (melting curve) analysis. When more than one amplicon was detected, primer sets were re-designed using the reported template sequence [20] and Primer-BLAST [68].  A single amplicon from these new primer sets were confirmed on a dissociation curve.”

The footnote of Table S3 has been modified to read “* The primers reported by Wang et al (2015) revealed multiple peaks (amplicons) using a dissociation curve (melting curve) analysis.  Therefore, primers for these genes were redesigned using the same template sequence reported by Wang et al. (2015).  These redesigned primer sets produced a single amplicon in the dissociation curve analyses. “

Reviewer 2 Report

This study investigated gene expression of four genes, one ABA biosynthetic gene, one ABA catabolic gene, and two ABA signaling pathway genes, in both early- and mid-season varieties of sweet cherry 5 days after GA3 treatment.

The Introduction contains too much information from review articles (lines 47, 49, 53, 55). It should be focused on fruit development and ripening of sweet cherry. For example, describe the definition of developmental stages of sweet cherry fruit.

The experiment design is not complete. The authors only investigated transcript abundance before and 5 days after GA3 treatment. How about the change of endogenous ABA? How about the gene expression during a time course, for example, T0, T1, T3, T5, T7…. How about the effects of GA3 on other ABA biosynthetic, catabolic and signaling genes?

According to the Materials and Methods, GA3 was applied at the light green-to-straw yellow transition stage of each variety, 36 and 29 DAFB in early-season variety Glendred and mid-season variety Lapins, respectively. But in Figure 1, fruit color of early-season Glendred 36 DAFB was yellow already, and the fruit color of mid-season Lapins 29 DAFB remains light green till 40 DAFB. Both descriptions are not consistent and will make readers confuse.

Author Response

Point 2.1: This study investigated gene expression of four genes, one ABA biosynthetic gene, one ABA catabolic gene, and two ABA signaling pathway genes, in both early- and mid-season varieties of sweet cherry 5 days after GAtreatment.

The Introduction contains too much information from review articles (lines 47, 49, 53, 55). It should be focused on fruit development and ripening of sweet cherry. For example, describe the definition of the developmental stages of sweet cherry fruit.

Response 2.1:  We would like to thank Reviewer 2 for their revision of the manuscript and comments. The modified version based upon this reviewer's comments as well as the other reviewers has been incorporated in the attached manuscript. 

We would like to clarify some points that this reviewer has made.  The reviewer is correct that we analyzed two ABA homeostasis genes, one associated with the biosynthetic pathway (PavNCED1) and the other associated with the catabolic pathway (PavCYP707A2) in the early and mid-season varieties of sweet cherry 5 days after GA3 treatment.  However, we analyzed the expression of four putative ABA response pathway orthologs (PavPP2C3, PavPP2C4, PavSNRK2.1 and PavSNRK2.3) as well.  In the early variety Glenred, we detected a decrease in transcript levels of PavPP2C3, PavPP2C4, PavSNRK2.3) 5 days after GA3 treatment, but no significant variations in relative transcript abundance was detected for the PavSnRK2.1  gene.  In contrast, in the mid-season variety Lapins, we detected a significant increase in PavPP2C3 and PavPP2C4 transcript levels (the opposite of what was seen in the early variety), but no significant variations in PavSNRK2.3 or PavSNRK2.1 relative transcript abundance.

We respectfully disagree with Reviewer 2 concerning the mention of review articles associated with ripening in non-climateric fruits.  The information presented in these lines and the review articles cited are important to contextualize the work and a source of references for the more general audience that reads Plants.  We have, however incorporated a more detailed description of sweet cherry fruit development as recommended by Reviewer 2.  Lines 63-67 now state: “Sweet cherry is a non-climacteric fleshy fruit plant with a typical double-sigmoid growth pattern characterized by: fast growth occurring at Stage I; slow growth of the pericarp together with embryo development and endocarp lignification leading to pit hardening in Stage II; and growth resumption and ripening initiation at Stage III [31-34]. During development of the sweet cherry fruit, there is a transition from green fruits to yellow, straw-yellow, pink and red. However, unlike other non-climateric fruit species, Prunus avium has a very short fruit ripening period. Sweet cherry color initiation and sugar increase at the onset of ripening correlate with ABA accumulation, which is known to trigger ripening [10,13,35].”

Point 2.2:  The experiment design is not complete. The authors only investigated transcript abundance before and 5 days after GA3 treatment. How about the change of endogenous ABA? How about the gene expression during a time course, for example, T0, T1, T3, T5, T7…. How about the effects of GA3 on other ABA biosynthetic, catabolic and signaling genes?

Response 2.2:  Although a time-course analysis of fruit development and further analyses of additional ABA biosynthetic, catabolic, and signaling genes are needed to future understand the association between the physiological effects of GA3 application under field conditions and the complex cross-talk between GA3 and ABA during sweet cherry fruit development,  as mentioned by Reviewers 1 and 4, the data presented in this work is novel and demonstrates that at the physiological and molecular level early and mid-season varieties of sweet cherry are responding differently at a physiological and molecular level.  The work presented in this paper sets the stage such that more complex studies may be able to reveal in time and development of sweet cherry fruits and how endogenous genotype (early vs. mid varieties) and exogenous factors (exogenous application of GA3) alter fruit ripening parameters and fruit quality.

Point 2.3  According to the Materials and Methods, GA3 was applied at the light green-to-straw yellow transition stage of each variety, 36 and 29 DAFB in early-season variety Glendred and mid-season variety Lapins, respectively. But in Figure 1, fruit color of early-season Glendred 36 DAFB was yellow already, and the fruit color of mid-season Lapins 29 DAFB remains light green till 40 DAFB. Both descriptions are not consistent and will make readers confuse.

Response 2.3: Thank you for this comment and we apologize for this confusion.  Based upon this comment, we realized that the color bar in Figure 1 (which is a schematic representation of the fruit colors) doesn’t represent the phenological data for these fruits in Table S1.  In this table, you can see that Glenred at 36 DAFB was 40% yellow and 60% green.  We have modified the color bar of Figure 1 to schematically represent fruit color more clearly.  Additionally, on line 109 and in the figure legend of Figure 1, we now make reference to Tables S1 and S2 where the color of the fruits at different stages of development are depicted in more detail.

Reviewer 3 Report

Although the attempt to understand the reported varietal differences in the response of sweet cherries to orchard GA application via its interaction with ABA is reasonable, this study is only preliminary and not quite ready for publication, leaving much unexplained. For example, the early Glenred hardly responds to GA compared to the significant Lapins response, yet the response of the two ABA regulatory genes examined is the reverse. Morover, the differences in the transcript accumulation of the ABA signal transduction genes do not supply a succint explanation. Possibly, an assessment of the ABA content of the fruit during ripening might clarify this point. As it is, the hypotheses presented in the discussion need to be supported by more data.

Specific comments:

Table 1 - GA significantly reduced the size of Lapins and increased that of Glenred. Is this not worthy of mention?  

line 217 -Suggestion: 'delayed softening' instead of 'increased firmness'.

line 218 - The quote from Choi et al. is not clear. They found no effect of GA on the PG and Cx activities in the early cvs. However, in this case the Glenred treated fruit is firmer than the control.

line 360 - What is the meaning of 'pericarp-enriched'?

Reviewer 4 Report

The aim of this work was to determine if GA3 modulates fruit ripening in two sweet cherry varieties having contrasting maturity time phenotypes. This was related to changes in the transcript abundance of ABA pathway related genes. Authors found that both varieties were responsive to GA3 and this was accompanied with changes in the transcript abundance of ABA homeostasis and signal transduction related genes. I highly recommend the acceptance of this MS which contains new insights of sweet sherry fruit ripening. Excellent!

Author Response

Point 4.1 The aim of this work was to determine if GA3 modulates fruit ripening in two sweet cherry varieties having contrasting maturity time phenotypes. This was related to changes in the transcript abundance of ABA pathway related genes. Authors found that both varieties were responsive to GA3 and this was accompanied with changes in the transcript abundance of ABA homeostasis and signal transduction related genes. I highly recommend the acceptance of this MS which contains new insights of sweet cherry fruit ripening. Excellent!

Response 4.1:  We would like to thank Reviewer 4 for their revision of the manuscript.  The modified version based upon the reviewers has been incorporated in the attached manuscript. 

Round 2

Reviewer 2 Report

If the authors want to clarify the different regulation mechanism between both varieties, the endogenous ABA with or without GA3 treatment should be measured. Abundance of transcript cannot reveal the level of bioactive ABA.

Reviewer 3 Report

Thank you for your response to my critique, which has convinced me to recommend publication following your revision.

Author Response

We would like to thank reviewer 3 for reviewing the modified version of our manuscript.